METHODS AND RESOURCES

# HCNetlas: A reference database of human cell type-specific gene networks to aid disease genetic analyses

**Jiwon Yu**[1]☯**, Junha Cha**[1]☯**, Geon Koh**[1]**, Insuk Lee** [1,2]*

**1** Department of Biotechnology, College of Life Science and Biotechnology, Yonsei University, Seoul, Republic of Korea, **2** POSTECH Biotech Center, Pohang University of Science and Technology (POSTECH), Pohang, Republic of Korea

☯ These authors contributed equally to this work.
* insuklee@yonsei.ac.kr

**Data Availability Statement:** The edge information of CGNs for HCNetlas and codes for the presented network analysis are freely available from https://github.com/netbiolab/HCNetlas. Source data and original codes used to generate all the figures in the

## Abstract

Cell type-specific actions of disease genes add a significant layer of complexity to the genetic architecture underlying diseases, obscuring our understanding of disease mechanisms. Single-cell omics have revealed the functional roles of genes at the cellular level, identifying cell types critical for disease progression. Often, a gene impact on disease through its altered network within specific cell types, rather than mere changes in expression levels. To explore the cell type-specific roles of disease genes, we developed HCNetlas (human cell network atlas), a resource cataloging cell type-specific gene networks (CGNs) for various healthy tissue cells. We also devised 3 network analysis methods to investigate cell type-specific functions of disease genes. These methods involve comparing HCNetlas CGNs with those derived from disease-affected tissue samples. These methods find that systemic lupus erythematosus genes predominantly function in myeloid cells, and Alzheimer's disease genes mainly play roles in inhibitory and excitatory neurons. Additionally, they suggest that many lung cancer-related genes may exert their roles in immune cells. These findings suggest that HCNetlas has the potential to link disease-associated genes to cell types of action, facilitating development of cell type-resolved diagnostics and therapeutic strategies for complex human diseases.

## Introduction

Human tissues comprise a mosaic of cell types, each with distinctive functional roles that affect how genes associated with diseases contribute to their onset and progression. Grasping how specific cell types influence the action of disease-related genes is a complex and as-yet unresolved aspect of human genetics [1,2]. The Human Cell Atlas (HCA) project [3], which compiles extensive single-cell RNA-sequencing (scRNA-seq) data from healthy tissues, seeks to illuminate the relationship between cell types and disease through gene expression profiles [4].

However, the role of a disease gene within a specific cell type often extends beyond expression levels to its position and influence within a gene network, known as network centrality.

manuscript is also available at https://doi.org/10.5281/zenodo.14522296.

**Funding:** This study was supported by the National Research Foundation funded by the Ministry of Science and ICT (2018R1A5A2025079, 2022M3A9F3016364, 2022R1A2C1092062 to IL). This work was supported in part by the Brain Korea 21 (BK21) FOUR Program (to JY, JC, and GK). The funders had no role in study design, data collection and analysis, decision to publish, or preparation of the manuscript.

**Competing interests:** The authors have declared that no competing interests exist.

**Abbreviations:** AD, Alzheimer's disease; AUROC, area under the ROC curve; CGN, cell type-specific gene network; DEG, differentially expressed gene; FDR, false discovery rate; GSEA, gene set enrichment analysis; HCA, Human Cell Atlas; HCNetlas, human cell network atlas; GOBP, Gene Ontology biological process; GSVA, gene set variation analysis; ICM, immune checkpoint molecule; IFN, interferon; ILC, innate lymphoid cell; ISG, interferon-stimulated gene; MTG, middle temporal gyrus; NK, natural killer; PBMC, peripheral blood mononuclear cell; PCA, principal component analysis; ROC, receiver operating characteristic; scRNA-seq, single-cell RNA-sequencing; SLE, systemic lupus erythematosus; SLEDAI, Systemic Lupus Erythematosus Disease Activity Index; snRNA-seq, single-nucleus RNA-sequencing; ssGSEA, single-sample gene set enrichment analysis; TCGA, The Cancer Genome Atlas; UMAP, uniform manifold approximation and projection.

To address this, we need a network-focused approach for mapping disease genes to their functional landscapes within specific cell types. In response, we have crafted scHumanNet [5], leveraging HumanNet [6] as the foundational interactome, refining connections based on cell-to-cell variation in gene expression [7]. This framework allows us to discern the network topologies of disease genes by contrasting cell type-specific gene networks (CGNs) [8] from healthy versus diseased tissues, leading to the identification of the cell types wherein disease genes are influential. Typically, this involves creating CGNs for both healthy and diseased tissue samples.

Identifying altered cellular states in diseases typically requires comparisons with matched control samples, which can entail additional costs, efforts, and sometimes may even be unavailable. This challenge can be mitigated if there is access to a comprehensive collection of reference cells from healthy individuals, like a cell atlas. Such a resource could potentially eliminate the need to generate control samples. In a similar vein, a network atlas that offers reference CGNs for a diverse array of cell and tissue types from healthy individuals could significantly streamline the investigation of disease-state network alterations. Moreover, while individual gene expression values are prone to aggregate according technical origins across different batches confounding biological variations, the inference of gene associations with adequate cell number is generally less affected by such variances especially when large number of samples are used (e.g., cells) [9]. This is because co-expression signals are intrinsically normalized within each batch, making them more reliable for network comparison. Therefore, we propose that an integrated collection of CGNs, derived from a cell atlas, would constitute a robust framework for cell type-specific analysis of disease genes. This approach would circumvent the need for matched control samples, offering a more efficient route to understanding disease mechanisms.

Here, we introduce HCNetlas (human cell network atlas), a collection of reference CGNs from a wide array of healthy tissue cells, which parallels the HCA in its potential impact on disease research. By utilizing these reference CGNs, it becomes feasible to uncover associations between disease genes and specific cell types, relying solely on the availability of disease sample data. This approach also bypasses the necessity to infer CGNs from matched control samples, streamlining the process of identifying disease-specific gene interactions within a specific cell type. HCNetlas currently includes 198 CGNs covering 61 cell types across 25 tissues (Table 1). We clustered the CGNs based on their disease profiles and observed the formation of groups comprising similar cell types. This clustering pattern indicates the potential of these CGNs to effectively resolve the cell type specificity of disease gene functions. Additionally, we implemented 3 network-based methods for assessing the cell type-specific functions of disease genes. Utilizing this analytical framework on both reference CGNs and those derived from diseased tissues enabled us to pinpoint cell types implicated in various diseases, thereby validating the effectiveness of our approach in cell type-resolved disease genetics. Consequently, HCNetlas holds great promise in expediting the discovery of biomarkers and therapeutic targets that are specifically tailored to the cellular context of disease genes, offering a refined perspective on the intricate web of cell type-specific gene actions in human diseases.

## Methods

### Single-cell and single-nucleus transcriptomic data used for HCNetlas construction

We employed both single-cell and single-nucleus RNA sequencing (scRNA-seq and snRNA-seq) data from multiple sources to reconstruct HCNetlas CGNs. We acquired scRNA-seq data

**Table 1. Tissue abbreviations used in this study.**

| Abbreviation | Tissue |
|---|---|
| BLD | Blood |
| BLR | Bladder |
| BMA | Bone marrow |
| FAT | Fat |
| JEJEPI | Jejunum epithelial |
| JEJLP | Jejunum lamina propria |
| LARITN | Large intestine |
| LIV | Liver |
| LLN | Lung-draining lymph nodes |
| LN | Lymph nodes |
| LNG | Lung |
| MG | Mammary |
| MLN | Mesenteric lymph nodes |
| MM | Muscle |
| PNC | Pancreas |
| PRS | Prostate |
| SG | Salivary gland |
| SKN | Skin |
| SMITN | Small intestine |
| SPL | Spleen |
| THY | Thymus |
| TRA | Trachea |
| VAS | Vasculature |
| M1 | Primary motor cortex |
| MTG | Middle temporal gyrus |

for 329,762 immune cells spanning 16 tissues from 12 deceased donors in a cross-tissue immune cell atlas [10]. We used the author-provided cell type labels, which were initially annotated by CellTypist and subsequently underwent manual curation. Additionally, we integrated data from Tabula Sapiens [11], which included the human transcriptome reference for 249,961 immune cells across 24 tissues from 15 donors. For constructing brain CGNs, we utilized snRNA-seq data of Allen Brain Atlas (http://www.brain-map.org) [12], derived from 76,533 nuclei in the primary motor cortex (M1) and 166,868 nuclei in the middle temporal gyrus (MTG) using 10× Genomics Chromium platform. All data were processed through alignment, quantification by Cell Ranger, and cell type annotation.

We harmonized pre-annotated cell types across all data sets and applied scHumanNet (v. 1.0.0) [5] to the single-cell transcriptomic data to generate CGNs for various tissues and cell types. Briefly, scHumanNet takes a single-cell transcriptome count matrix for a specific cell type, followed by data imputation, transformation, and normalization. It then produces a transformed gene activity score matrix and filters the reference interactome, HumanNet, for gene pairs in the matrix that pass a minimum activity score, creating a CGN. The edge weight scores are inherited from the reference interactome. scHumanNet has been shown to be a superior network inference model compared to other single-cell based methods. To construct CGNs for major cell types, we aggregated sub-cell types into broader categories, such as T cells, B cells, and myeloid cells. In total, 198 CGNs were generated, encompassing 25 tissues and 61 cell types.

## Evaluating the cell type-specificity of CGNs

We performed dimension reduction analysis with uniform manifold approximation and projection (UMAP) to visualize interrelationships among HCNetlas CGNs. Utilizing scHuman-Net for CGN construction, which references the HumanNet [6] comprising 18,593 human genes, we generated binary profile vectors indicating the presence (1) or absence (0) of each gene in the network for every CGN. For visualizations, we employed the UMAP package in R (v. 0.2.10), setting min_dist to 0.5 to balance the trade-off between local and global structure in the data.

To determine the cell type-specific functionality of the HCNetlas CGNs, we explored the enrichment of cell type-associated genes, particularly for B cells and T cells. We collated cell type-associated genes from 3 authoritative databases: Gene Ontology biological process (GOBP), CellMarker, and the Azimuth cell type database [13–15]. We postulated that genes functionally connected within a CGN reflect the properties of their respective cell type, thus we considered the interconnectivity within genes for each cell type as a measure of the cell type specificity.

Additionally, we identified hub genes within each CGN, which are likely to play pivotal roles in the function of their corresponding cell type. We profiled the top 15 hub genes, ranked by degree centrality, for each major cell type across various tissues using the *FindAllHub()* function of the scHumanNet. *FindAllHub()* function randomly shuffles the constructed network and creates a null distribution of centrality values. This enables the calculation of statistical significance for the hubness of a gene of interest. This profiling helped to ascertain the relative importance of these hub genes within the network.

## Overall assessment of the association between CGNs and diseases

We assessed whether HCNetlas CGNs can discern connections between diseases and cell types by profiling CGNs with disease-association scores. Within each CGN, we ranked the 18,593 genes from the HumanNet reference interactome by degree centrality, utilizing the *GetCentrality()* function of the scHumanNet. Disease gene sets, totaling 5,763, were compiled from DisGeNET [16] and GWAS Catalog [17] for the analysis. Assessment of disease gene set association with each CGN was performed with ssGSEA (v. 2.0) [18] and GSVA [19], the latter via the *gsva()* function from the GSVA package (v. 1.38.2). Hierarchical clustering of disease gene sets and CGNs was performed using the R package ComplexHeatmap, with the default parameters, where cluster.rows and cluster.cols were both set to TRUE.

Furthermore, we conducted differential compactness analysis on the HCNetlas CGNs using the same disease gene sets. For each gene set, we calculated the within-group connectivity across all 198 networks to gauge network compactness. To accommodate variations in network size, we normalized the within-group connectivity by the number of nodes in each network and then scaled these normalized values by multiplying them by 10,000 to ensure a consistent basis for comparison across all networks.

## Cell type-resolved genetic analysis of systemic lupus erythematosus (SLE) with HCNetlas

We acquired scRNA-seq data for peripheral blood mononuclear cells (PBMCs) from 41 patients with SLE and 15 healthy controls as reported by Nehar-Belaid and colleagues [20]. To ensure a consistent analysis, we excluded 2 SLE patient samples lacking SLEDAI scores. Following quality control measures, including the removal of doublets using the DoubletFinder (v. 2.0.0) [21] package and the exclusion of cells with fewer than 400 transcripts or

over 5% mitochondrial gene content, the data set was narrowed to approximately 276,000 cells. After normalization and scaling with Seurat package (v. 4.1.1), we identified 3,000 variable genes using the *vst()* and *FindVariableFeatures()* functions. Batch effects were mitigated by applying principal component analysis (PCA) and Harmony (v. 1.0) [22] (dims = 40), and cellular clustering was performed using the Louvain method (resolution = 1.5), followed by UMAP visualization with 40 dimensions. Cell types were manually annotated using canonical markers after optimizing the number of principal components and clustering resolution.

For constructing SLE-specific CGNs, we focused on the curated data from SLE patients. We built networks for B cells, CD4$^+$ T cells, CD8$^+$ T cells, myeloid cells, and NK cells. Using the *Compactness()* function, we performed differential compactness analysis. We referenced 184 SLE-associated genes from KEGG pathway (I05322) and KEGG disease (H00080) databases, comparing connectivity within disease CGNs and HCNetlas CGNs, and visualized the networks in Cytoscape (v 3.9.1) [23].

Node centrality within these networks was computed using the *GetCentrality()* function from the scHumanNet package. We compared the percentile ranks of centrality for disease CGNs against the reference CGNs of HCNetlas using the *DiffPR.HCNetlas()* function. Genes showing differential hubness were pinpointed with *FindDiffHub.HCNetlas()*, with significance defined by a *q*-value < 0.05 after Benjamini–Hochberg correction to control false discovery rate (FDR).

We compiled interferon-stimulated genes (ISGs) from hallmark gene sets of the molecular signature database (MSigDB) and the Immunological Genome Project (ImmGen) [24], resulting in a total of 423 ISGs. The efficacy of prediction for ISGs by hubness within SLE-associated CGNs was assessed using receiver operating characteristic (ROC) curve analysis. The ROC curve was generated using the *roc()* function from the pROC package (v. 1.18.0).

To explore the diagnostic potential of gain-of-hubness genes, we computed an expression score for the genes in myeloid cells via *AddModuleScore()* of the Seurat package and evaluated the differences in distribution of expression values between patients and healthy controls using the Wilcoxon signed-rank test.

Differentially expressed gene (DEG) analysis was performed by merging Seurat objects containing HCNetlas healthy tissue data with disease scRNA-seq data. After normalization and scaling by a factor of 10,000, we identified 2,000 variable genes. DEGs were pinpointed for key cell types using Seurat's *FindMarkers()* function, considering genes with an adjusted *p*-value < 0.05 and an absolute log$_2$-fold change > 0.5, focusing solely on coding genes.

## Cell type-resolved genetic analysis of Alzheimer's disease (AD) with HCNetlas

In our study of AD, we used snRNA-seq data from 12 patients with annotated cell types from Morabito and colleagues [25]. Since the data were derived from the prefrontal cortex of brain tissues, the generated CGNs for AD were compared with reference CGNs for the primary motor cortex (M1) from the HCNetlas. We grouped the cell type annotations into 4 main categories: astrocytes, inhibitory neurons, excitatory neurons, and oligodendrocytes. The identification of differential hubness genes and DEGs within these cell types was carried out using the same methodology applied in the analyses of SLE. To ascertain the relevance of AD-associated genes predicted by our differential hubness analysis, we referenced genes linked to AD in the KEGG pathway (M16024), MSigDB (M35868), and Wightman and colleagues [26].

## Evaluation of differentially associated pathways between reference and disease CGNs

Considering the association of gain-of-hubness and loss-of-hubness genes with AD in inhibitory and excitatory neurons, we constructed network-ranked signatures for both reference and AD-specific CGNs for the cell types. The signature genes were based on the top 10 hub genes by degree centrality within each CGN. The networks of these top-tier hub genes were visualized using the Cytoscape software [23]. Furthermore, we conducted gene set enrichment analysis (GSEA) on these network-ranked signatures using the enrichR package [27]. To evaluate the pathways differentially associated between disease CGNs and reference CGNs, we introduced a metric called *diffQ*, calculated as follows:

$$diffQ = -\log_{10}\left(\frac{q - \text{value of association with disease CGN}}{q - \text{value of association with reference CGN}}\right)$$

In this formula, a positive *diffQ* value signifies that a pathway is more strongly associated with the cell type in its diseased state than in its healthy state (gain-of-pathway). Conversely, a negative *diffQ* value indicates greater association with the cell type in its healthy state as compared to its diseased state (loss-of-pathway). To emphasize the most significantly altered pathways, we focused on the top 10 KEGG pathways with the highest absolute *diffQ* values. This approach effectively pinpoints the key molecular pathways involved in the pathogenesis of AD.

## Cell type-resolved genetic analysis of lung cancer using HCNetlas

To create lung cancer-specific CGNs, we used scRNA-seq data from 29 tumor tissues provided by Qian and colleagues [28]. We retained the pre-annotated cell type identifications from the data sets. For comparison with reference CGNs derived from paired normal tissues, we constructed networks from both the lung cancer and healthy control data. Following the scHumanNet protocol, we generated networks and defined differential hubness genes using *FindDiffHub()*. The process of identifying differential hubness genes within each cell type was conducted using the same methodology employed in the SLE analyses. Similarly, the identification of DEGs followed the methodology used in the SLE studies, with the exception that genes exhibiting an absolute $\log_2$-fold change $< -1.5$ were categorized as down-regulated DEGs.

To validate the lung cancer relevance of the identified genes, we referenced the Cancer Gene Census, CancerMine, and IntOGen databases [29–31]. We assessed the proportion of lung cancer-associated genes detected uniquely through differential hubness, uniquely through DEGs, and by the intersection of both methods. Furthermore, we analyzed 42 immune checkpoint molecules listed by Auslander and collagues [32] to determine if cell type-specific genes vital for cancer immunity are discernible through both expression-based and network-based analyses.

We investigated the prognostic potential of genes identified by cell type-specific differential hubness and differential expression using survival analysis on TCGA lung cancer data sets (TCGA-LUSC, TCGA-LUAD). Initially, we identified a total of 379 gain-of-hubness genes and 211 up-regulated DEGs from 3 major cell types: B cells, T cells, and myeloid cells. Subsequently, genomic and clinical data for 1,017 lung cancer samples were acquired from the GDC portal [33]. The STAR-Counts data underwent preprocessing, log-normalization, and variance stabilization using the *vst()* function in the DESeq2 R package (v. 1.30.1). With the application of GSVA [19], we evaluated the association of both gain-of-hubness genes and up-regulated DEGs with each tumor expression profile of the patients. Patients were then classified into upper and lower quartile groups based on their GSVA scores. These groups were further

examined through Kaplan–Meier survival curves. To ensure the reliability of our findings, we adjusted all *p*-values obtained from the survival analysis using the Benjamini–Hochberg method to control the FDR.

## Results

### HCNetlas: A catalog of reference CGNs for various healthy human tissues

To build reference CGNs, we utilized scRNA-seq data from immune cell atlas project [10] and The Tabula Sapiens [11], as well as single-nucleus RNA-sequencing (snRNA-seq) data from the Allen Brain Atlas [12]. Our single-cell transcriptomic data set comprised 763,559 cells from 28 donors. We generated gene networks for each predefined cell type using the scHumanNet framework [5] (Fig 1A), providing a comprehensive baseline for identifying disease-associated genes and cell types.

When constructing CGNs, the number of cells used can influence the efficacy of network inference. To investigate this aspect, we conducted an analysis of CGNs derived from various cell atlas data sets, specifically examining how the number of cells used for network inference correlates with the overall network size. Our findings indicated a clear trend: as the number of cells increases, there is a corresponding rise in both the node and edge counts within the inferred CGNs. However, this growth in network complexity tends to plateau once the cell count reaches approximately 1,000 (S1A Fig). The observed saturation point suggests that the inference of CGNs becomes substantially robust to the effects of sample size when the number of cells exceeds 1,000. Based on this insight, we focused our study on networks inferred from data sets comprising a minimum of 1,000 cells. This led to the generation of 198 CGNs, covering 25 tissues and including 61 distinct cell types (S1 Table). These networks form our newly established resource, HCNetlas, a catalog of human CGNs for healthy tissues.

To examine the interrelationships among the CGNs in our HCNetlas, we analyzed each CGN based on network gene profiles, subsequently visualizing these profiles in a reduced dimensional space. This analysis demonstrated a clear trend where CGNs corresponding to the same cell types exhibited a tendency to cluster together (Fig 1B), reinforcing the concept that these networks accurately capture and reflect the specificity inherent to each cell type. Notably, CGNs within the myeloid and B cell lineages showed remarkable coherence, in contrast to the T cell lineage CGNs, which exhibited greater heterogeneity. An interesting observation was the close proximity of innate lymphoid cell (ILC) and natural killer (NK) CGNs (S1B Fig), underscoring their lineage correlations [34]. However, CGNs related to the same tissue types generally did not demonstrate strong clustering with the exception brain tissue network nodes that displayed high similarity (Fig 1C), suggesting that cell type identity is a stronger determinant of network structure than tissue environment. This was further evidenced in the T cell lineage, including ILCs, NK cells, CD4$^+$ T cells, and CD8$^+$ T cells, where subsets exhibited coherence within cell types but not necessarily within tissue types (S1B and S1C Fig). This aligns with recent studies that emphasize tissue or sub-cell type-dependent variability in T cells [35–37]. These findings highlight the utility of HCNetlas as a potentially powerful tool for investigating cell type-specific gene functions.

### Assessing the cell type-specific functionality of HCNetlas CGNs

To evaluate whether the reference CGNs of HCNetlas accurately reflect cell type-specific functions, we conducted tests using 2 distinct immune cell types from different lineages: B cells and T cells. The premise of this test was that if the HCNetlas CGNs are effective in representing cellular functions unique to each cell type, then genes for maintaining the identity and function of each cell type should demonstrate interconnectedness within their respective networks.

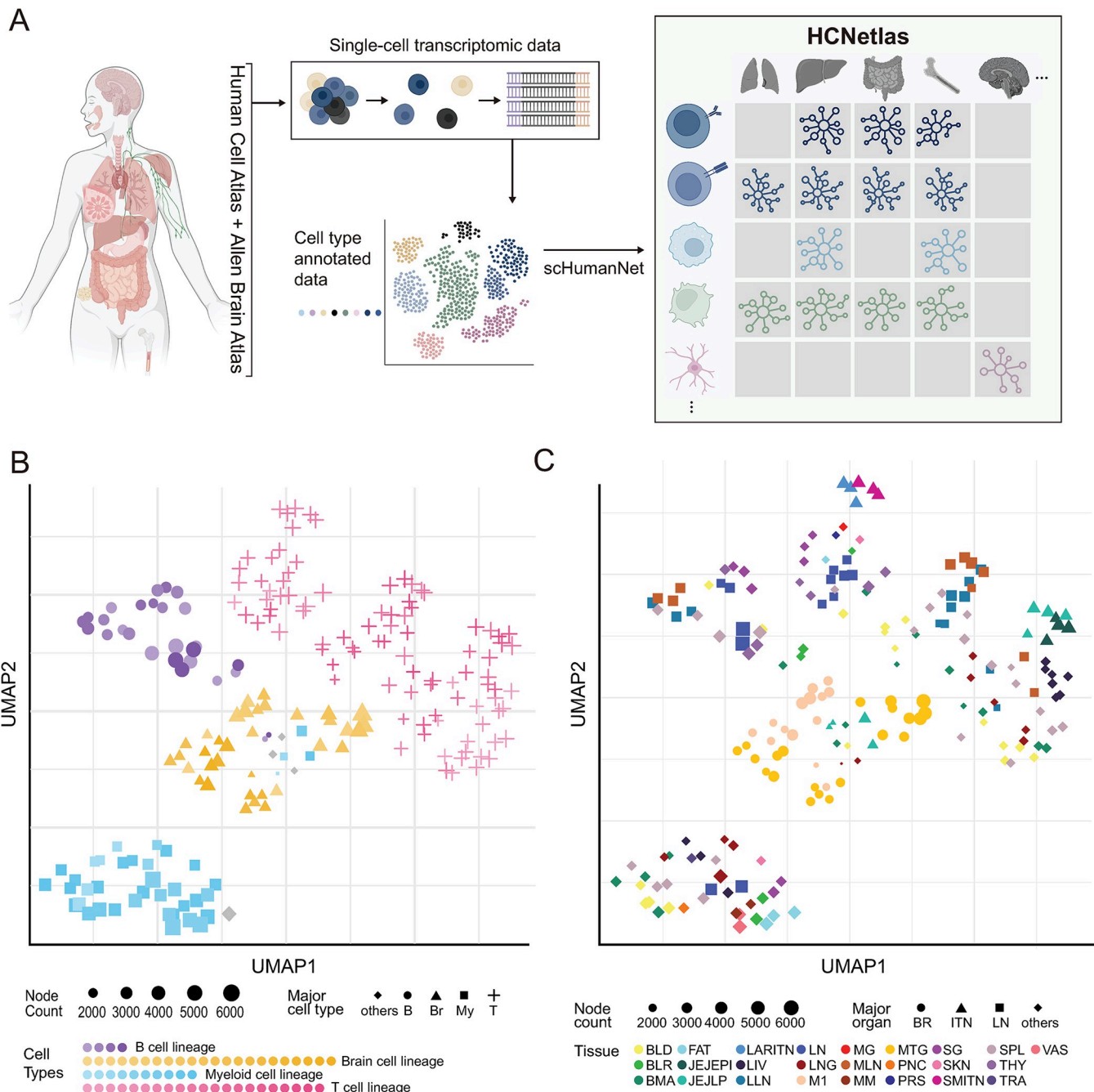

**Fig 1. Overview of HCNetlas.** (**A**) Schematic representation of the workflow from single-cell transcriptomic data collection to the construction of the HCNetlas. Single-cell RNA sequencing data preannotated for cell type were used to build CGNs using the scHumanNet framework. HCNetlas is comprised of a comprehensive collection of these gene networks, representing various human tissues and cell types. (**B**) UMAP visualization of CGNs based on gene profiles, highlighting the major cell lineages, with node size representing the number of genes in each network. Major cell type "Other" in gray (major cell type abbreviations; B; B cells, Br; Brain cells, My; Myeloid cells, T; T cells). (**C**) UMAP plot displaying the interrelationship among the CGNs based on network gene profiles for major organs or tissue types. Each point represents a gene network associated with a specific organ or tissue type colored distinctly. The data underlying this figure can be found in https://doi.org/10.5281/zenodo.14522296. CGN, cell type-specific gene network; HCNetlas, human cell network atlas; UMAP, uniform manifold approximation and projection.

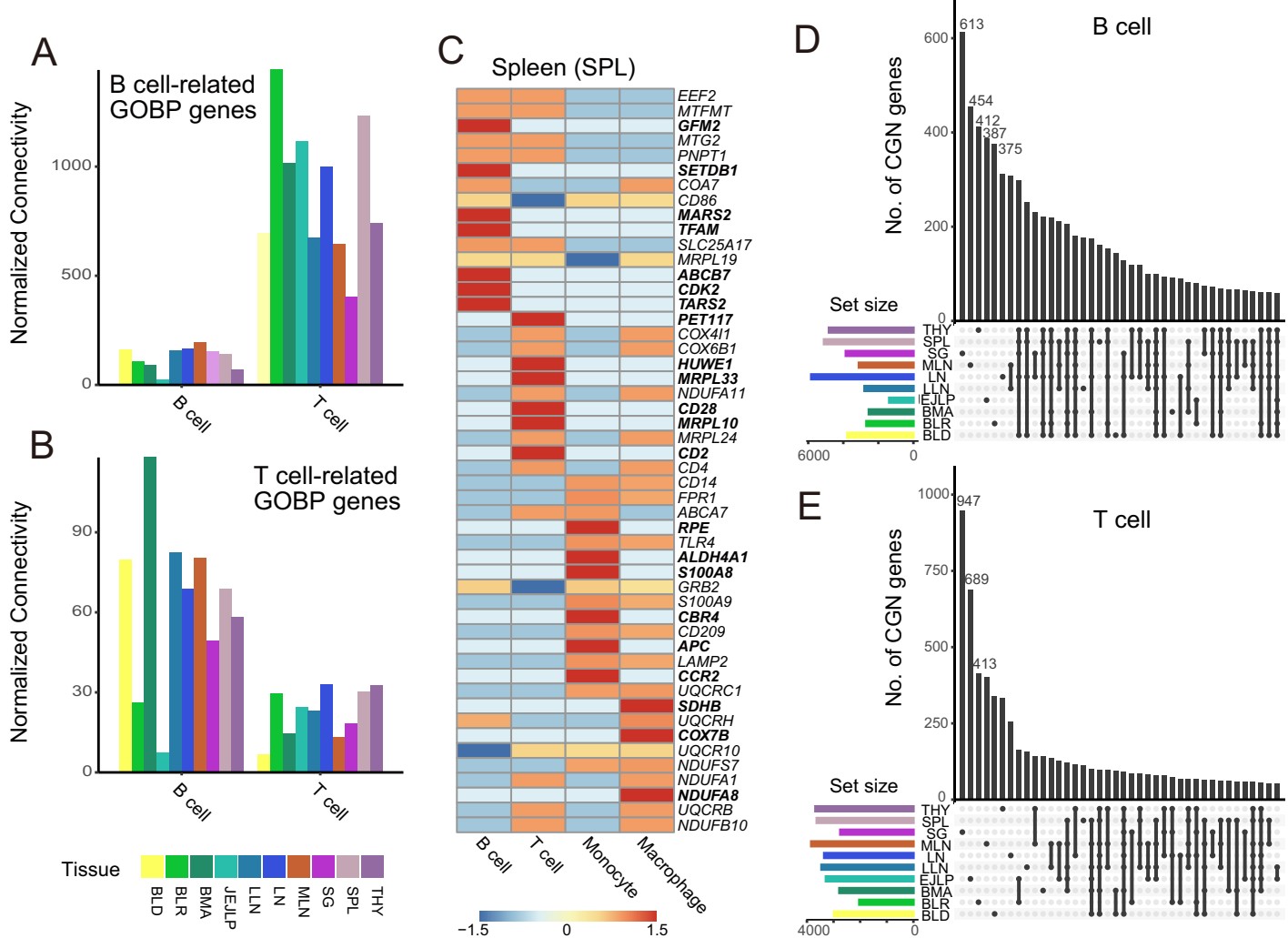

**Fig 2. Cell type-specific functionality of HCNetlas CGNs. (A, B)** Bar graph illustrating the within-group connectivity of B cell-related (**a**) or T cell-related (**b**) GOBP genes in the respective CGNs. Connectivity is normalized by each CGN's total node number. All tissues with over 0 value of normalized connectivity for both B and T cells are included. (**C**) Heatmap displaying the percentile rank of the top 15 hub genes in spleen CGNs, with values scaled per row, with color intensity indicating the expression level from low (blue) to high (red). Genes uniquely found in each cell type are highlighted in bold. (**D, E**) UpSet plots for 2 major CGNs, B cell CGN (**D**), and T cell CGN (**E**), representing the intersection of network genes across different tissues. The data underlying this figure can be found in https://doi.org/10.5281/zenodo.14522296.

As anticipated, our analysis showed that genes specifically annotated for either B cells or T cells by the GOBP [38] exhibited the highest within-group connectivity in their respective CGNs across various tissues (Fig 2A and 2B). This pattern of connectivity was further validated by comparing it with cell type marker genes as identified in the Azimuth database [13] and the CellMarker database [14] (S2A Fig). These findings underscore the ability of HCNetlas CGNs to capture and represent the unique functional characteristics inherent to specific cell types.

Moreover, we investigated the network hub genes within each CGN, identified based on degree centrality (Fig 2C). For instance, in spleen CGNs, *CD86*, which is pivotal in B cell activation [39], emerged as top hub genes in the B cell CGN. Similarly, genes essential for T cell identity like *CD2*, *CD4*, and *CD28* were among the top 15 hub genes in the T cell CGN [40].

Additionally, *S100A8*, *S100A9*, *CD14*, markers for monocytic myeloid-derived suppressor cells were prominent hubs in the monocyte CGNs [41]. These patterns of hub genes, significant due to their high degree centrality, were consistent across various tissues (S2B Fig), underlining the functional interpretability of these hub genes in the context of their respective cell types.

Lastly, to assess the tissue dependency of the HCNetlas CGNs, we compared CGN genes for each major cell type across different tissues. We observed limited overlap in CGN genes among different tissues within major cell types (Figs 2D, 2E, and S3), suggesting a convergence of networks across tissues within major cell lineages, aligning with findings from previous studies [10,35]. These observations indicate that while there are core gene networks characteristic of each cell type, tissue adaptation of CGNs is also evident, underscoring the complexity and diversity of cellular functions across different biological contexts.

## HCNetlas as a tool for unraveling cell type specificity of disease genes

The HCNetlas, with its collection of reference CGNs, presents a promising resource for dissecting the cellular specificity of disease genes. The majority of disease-associated genes identified to date have been derived from bulk tissue data, which often fails to specify the exact cell types involved in disease onset and progression. In this scenario, HCNetlas CGNs become instrumental in pinpointing the critical cell types at play. To ascertain the effectiveness of HCNetlas CGNs in disease-oriented research, we embarked on an investigation to determine if these CGNs capture and reflect the cell type specificity of various diseases. This involved conducting enrichment analyses on the CGNs using disease-associated genes sourced from 2 distinct databases: DisGeNET [16] and GWAS Catalog [17]. We began by ranking genes within each CGN based on network degree centrality, and then applied single-sample gene set enrichment analysis (ssGSEA) [18] and gene set variation analysis (GSVA) [19] to profile degree of association with each set of disease genes. Our analysis revealed a distinct pattern of congregation among CGNs corresponding to shared cell types, as determined by disease-association profiles (Figs 3A and S4A). This finding was particularly notable within cell types, whereas the convergence of networks corresponding to the same tissue types was less pronounced, indicating the specificity of cell types in the context of disease genetics.

We next evaluated the connectivity among genes associated with the same disease within CGNs across different tissues. Our hypothesis was that genes would exhibit more interconnectedness in the relevant cell types and tissues primarily responsible for diseases. This analysis aimed to elucidate the relationships between specific diseases and their associated cell types or tissues. While not all diseases we considered manifests cell type specificity, we noticed that CGNs predicted similar disease gene enrichment patterns in tissues such as the intestine and the liver (Fig 3A). A case in point is hepatitis-related terms, where genes associated with this condition showed the most significant within-group connectivity in liver CGNs of most major immune cell types (Fig 3B). Noteworthy was the pronounced within-group connectivity observed in both myeloid cell and T cell CGNs, highlighting the integral role of T cells in viral infectious diseases and the contribution of Kupffer cells (resident liver macrophages) to hepatitis [42]. This finding indicates that HCNetlas effectively identifies relevant cell types and tissues implicated in hepatitis. Furthermore, genes related to schizophrenia showed increased within-group connectivity across brain tissues, particularly in the primary motor cortex (M1) and MTG CGNs (S4B Fig). Additionally, AD-associated genes from the GWAS were found to be highly connected within the macrophage and myeloid CGNs (S4C Fig), consistent with the association between AD risk and macrophage transcriptional network [43]. These observations underscore the potential of HCNetlas CGNs as

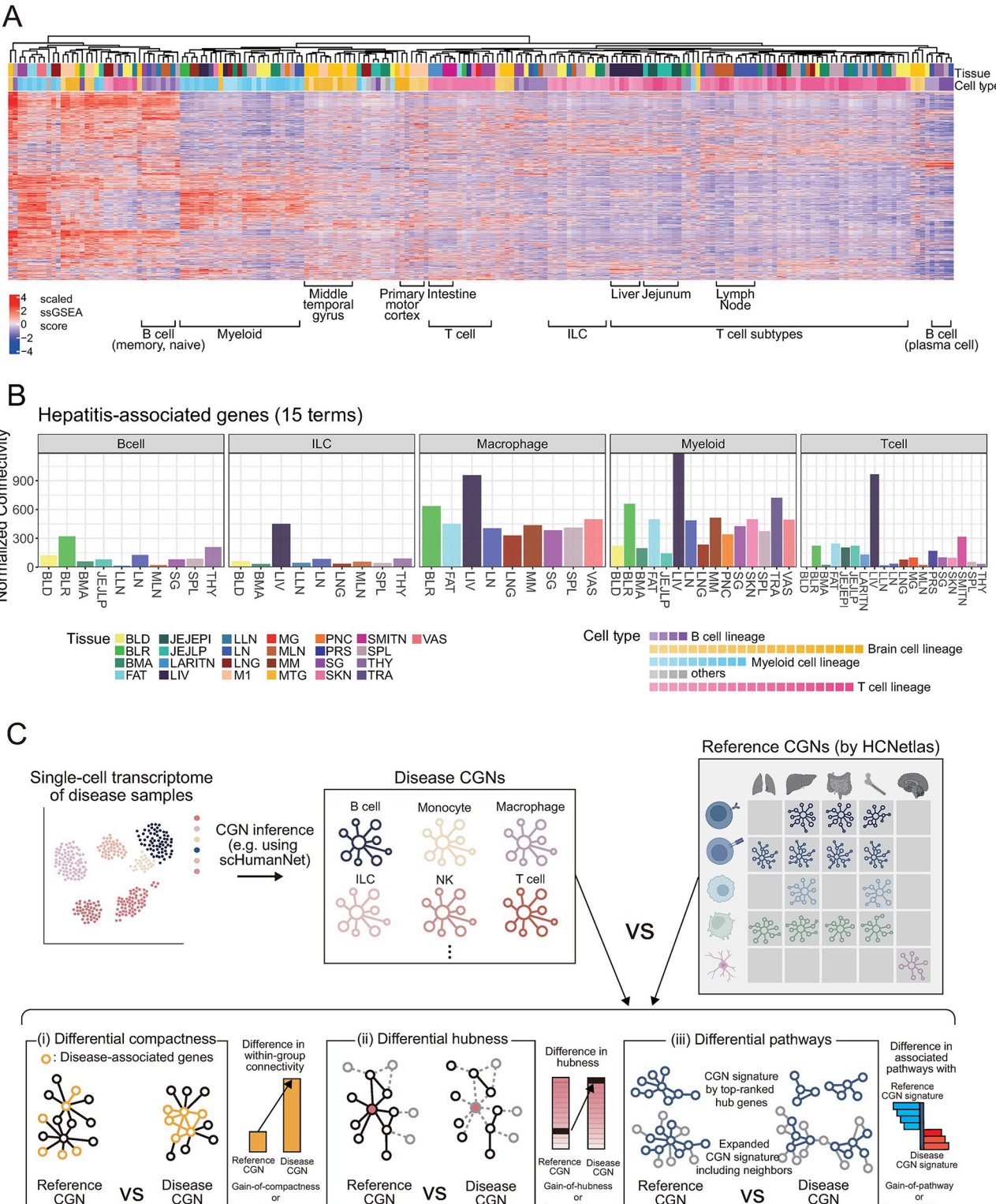

**Fig 3. Overview of cell type-resolved disease genetics using HCNetlas.** (**A**) Heatmap displaying the disease profiles of various cell types across different tissues, conducted with ssGSEA. Each column represents a CGN of HCNetlas, while each row corresponds to a disease gene set sourced from either DisGeNET or GWAS Catalog, ordered by hierarchical clustering. Color intensity indicates scaled ssGSEA enrichment values for the disease gene sets (rows), reflecting the degree of association of the CGN signature genes with each disease terms. (**B**) Bar graphs showing the within-group connectivity of genes associated with toxic hepatitis across different cell lineages, in various tissues. The bars are color-coded to represent different cell

lineages; 15 terms and their combined genes were assessed from DisGeNET terms based on the key word search "hepatitis." (**C**) Schematic representation and summary of the analytical framework used for comparing disease CGNs with reference CGNs from HCNetlas. The workflow illustrates the process of CGN inference from single-cell transcriptomes of disease samples and contrasts disease CGNs for specific immune cells against reference CGNs. The analysis includes (i) differential compactness, highlighting the difference in interconnectivity within disease-associated genes; (ii) differential hubness, showing the changes in hubness; and (iii) differential pathways, contrasting pathway associations between disease and healthy states based on enrichment for CGN signature genes. The data underlying this figure can be found in https://doi.org/10.5281/zenodo.14522296. CGN, cell type-specific gene network; HCNetlas, human cell network atlas; ssGSEA, single-sample gene set enrichment analysis.

a valuable resource for uncovering intricate relationships between diseases and specific cell types or tissues, thereby enhancing our understanding of disease pathology at a cellular level.

HCNetlas, having proven its functional and biological relevance, is posited to be an effective reference for network analyses in disease studies. To enhance their utility, we have developed a suite of network analysis methodologies (Fig 3C) and applied them to investigate various diseases, showcasing the adaptability of HCNetlas CGNs.

Firstly, if we have a set of disease genes, determining the specific cell type where these genes predominantly influence disease progression is crucial. To evaluate the functional role of these disease genes in a targeted cell type, we have developed an approach known as differential compactness analysis. This method compares the degree of interconnectivity among disease genes between the reference CGNs in HCNetlas and their corresponding disease CGNs derived from disease samples. In this framework, "gain-of-compactness" denotes an enhanced interconnectivity of disease genes within disease CGNs, suggesting an increased functional role in the disease context. Conversely, "loss-of-compactness" implies a reduced interconnectivity. Through this analysis, we can gain insights into which cell type the disease genes are actively involved and determine whether their impact on the disease state is characterized by a gain or loss of function.

Secondly, to identify disease genes and ascertain the cell type implicated in the disease, focusing on genes exhibiting significant differences in network centrality between diseased and healthy states can be insightful. Therefore, we prioritize genes based on differential hubness between disease CGNs and reference CGNs. This methodology involves categorizing genes into 2 distinct groups: "gain-of-hubness" and "loss-of-hubness." Genes in the "gain-of-hubness" category show increased centrality in disease CGNs compared to reference CGNs, indicating a heightened role in the disease state. Conversely, genes in the "loss-of-hubness" category demonstrate decreased centrality in disease CGNs, suggesting a reduced or altered function in the disease context. This approach effectively distinguishes genes that are central to disease mechanisms in specific cell types.

Lastly, examining pathways that show differential associations between diseased and healthy states in cell types associated with the disease can provide insights into the molecular mechanisms underlying pathogenesis. To conduct the differential pathway analysis, we initially select signature genes representative of both diseased and healthy states for each cell type. This selection is based on identifying the top-ranked hub genes (for example, the top 10 hub genes) within both the disease CGN and the corresponding reference CGN. Subsequently, through gene set enrichment analysis, we aim to identify and prioritize pathways that are differentially associated between the disease and healthy states. In this context, "gain-of-pathways" refers to those pathways that show an increased association with the disease state in comparison to the healthy control. Conversely, "loss-of-pathways" denotes pathways that have a reduced association in the disease state compared to the healthy state. Identifying these differentially associated pathways enables us to formulate hypotheses that delve deeper into the molecular basis of pathogenesis in disease-associated cell types.

## Cell type-resolved genetic analysis of an autoimmune disease using HCNetlas

Given that the majority of the CGNs provided by HCNetlas are derived from immune cells, this resource would be particularly valuable for studying immune disorders such as autoimmune diseases. To evaluate capability of HCNetlas to identify the specific immune cell types where disease genes have an impact on pathogenesis, we focused our research on SLE which is a chronic autoimmune disorder characterized by elusive pathogenesis, genetic susceptibility, and clinical heterogeneity [44]. For constructing disease CGNs for SLE, we manually annotated scRNA-seq data from 38 SLE patients [20] using canonical markers (Fig 4A). These disease CGNs were then compared with the blood cell CGNs from HCNetlas, providing insights into the cell type specificity underlying SLE pathogenesis.

Leveraging the principle that increased network compactness among disease-associated genes within a CGN indicates their significant role in the pathogenesis for that cell type, we assessed the involvement of major immune cell types in SLE. We applied a set of SLE-susceptible genes (S2 Table), gathered from the KEGG pathway database, to both disease CGNs and reference CGNs. Our analysis revealed that network compactness in myeloid cells and B cells is significantly greater in the disease CGN compared to the reference CGN (Fig 4B). This suggests that SLE-susceptible genes are critically involved in the disease progression primarily through myeloid cells and B cells.

Considering that genes associated with SLE predominantly exert their effects through myeloid cells, we prioritized genes for SLE based on network centrality within both disease and reference CGNs specifically pertaining to myeloid cells. Aligning with previous studies that emphasize the increased expression of type 1 interferon (IFN) and ISGs in SLE patients [20,45,46], we evaluate the prediction of SLE-associated genes based on retrieval rate of ISGs (S3 Table) using the ROC curve. Consistent with the greater network compactness of SLE-susceptible genes in the disease myeloid CGN relative to the reference myeloid CGN, our results showed a significantly improved prediction of ISGs in the disease myeloid CGN when compared to the reference myeloid CGN (Fig 4C). Likewise, for other cell types, disease CGNs demonstrated improved predictions of ISGs compared to the reference CGNs (Fig 4D and S4 Table). Taken together, these findings underscore the critical role of myeloid cells in the initiation and progression of SLE, corroborating previous research that highlights the link between SLE and myeloid cells [47–49].

Next, we hypothesized that gain-of-hubness genes for myeloid cells could effectively differentiate diseased myeloid cells from their healthy counterparts. To test this hypothesis, we initially identified a set of 131 gain-of-hubness genes with statistical significance (S5A Table). We then examined the distribution of their expression level between disease-state myeloid cells and their corresponding healthy controls. Our observations revealed a significant disparity between these 2 distributions, affirming the potential of these 131 gain-of-hubness genes to distinguish diseased myeloid cells (Fig 4E, left panel). Furthermore, we observed a positive correlation between the expression levels of these gain-of-hubness genes and the Systemic Lupus Erythematosus Disease Activity Index (SLEDAI) scores, albeit with limited statistical power due to the small sample size (Fig 4E, right panel). This correlation indicates that the expression patterns of these 131 gain-of-hubness genes are not only distinctive of diseased states but may also reflect the severity of SLE in patients. In contrast, the up-regulated DEGs in disease-state myeloid cells (S5B Table) did not demonstrate the capability to either differentiate diseased myeloid cells (Fig 4F, left panel) or correlate with SLEDAI scores (Fig 4F, right panel). These outcomes imply that collections of CGNs for healthy tissues, such as those provided by HCNetlas, are apt references for identifying disease states.

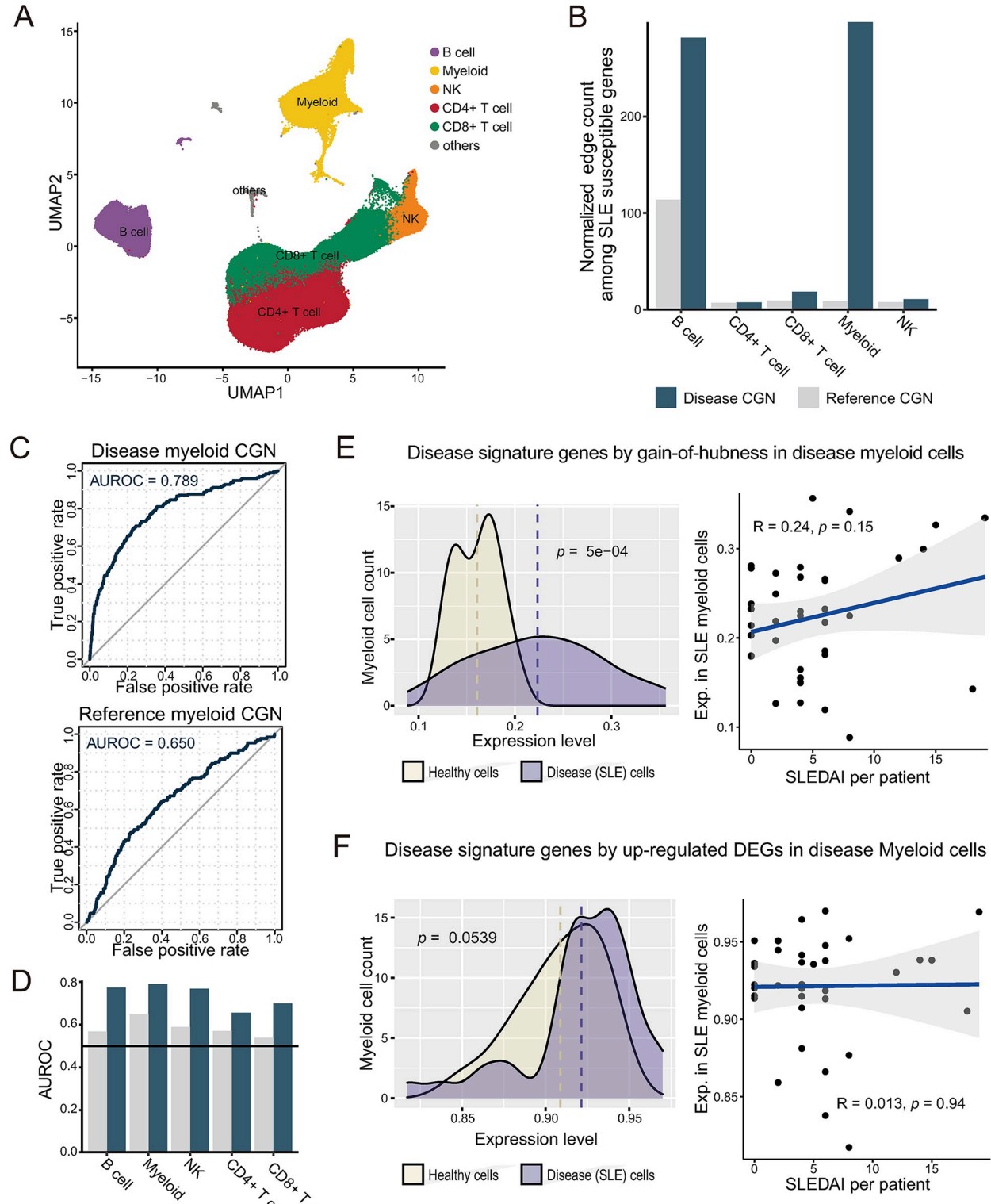

**Fig 4. Cell type-resolved disease genetics for SLE.** (**A**) UMAP plot representing the interrelationship among immune cells. (**B**) Bar chart showing normalized interconnectivity among SLE-susceptible genes in both reference and disease CGNs for various cell types. (**C**) ROC curves for retrieval of ISGs by network hubness in disease myeloid CGN and reference myeloid CGN. (**D**) Comparison of AUROC values with CGNs for various cell types, contrasting the reference and disease CGNs to assess prediction capability. (**E**) Left panel: Distribution of expression levels for 131 gain-of-hubness genes in myeloid cells. Right panels: Correlation analysis of expression levels of the 131 gain-of-hubness genes with the SLEDAI. (**F**) Same

as for (**E**) except using up-regulated DEGs. The data underlying this figure can be found in https://doi.org/10.5281/zenodo.14522296. AUROC, area under the ROC curve; CGN, cell type-specific gene network; DEG, differentially expressed gene; ISG, interferon-stimulated gene; ROC, receiver operating characteristic; SLE, systemic lupus erythematosus; SLEDAI, Systemic Lupus Erythematosus Disease Activity Index; UMAP, uniform manifold approximation and projection.

## Cell type-resolved genetic analysis of a brain disease using HCNetlas

HCNetlas offers an extensive collection of CGNs for brain tissue, making it a valuable resource for investigating neurological disorders. Alzheimer's disease (AD), a widespread neurodegenerative condition known for its progressive impact on behavior and cognitive functions, is one such area where HCNetlas can be particularly useful. To study AD more closely, we have developed disease CGNs using single-nucleus RNA sequencing (snRNA-seq) data from the prefrontal cortex of AD patients [25]. These disease CGNs were compared with HCNetlas CGNs, which were derived from the primary motor cortex (M1). This comparison enables a detailed analysis of alterations in the gene network that is associated with AD, facilitating a deeper understanding of the disease progression and its impact on brain function.

To identify the primary cell types impacted by AD-associated genes compiled from various sources (Methods, S6 Table), we employed differential compactness analysis. This analysis revealed that AD-associated genes exhibit a high degree of interconnectivity within the reference CGNs for both inhibitory and excitatory neurons (Fig 5A), suggesting that these genes predominantly function in these neuron types. Interestingly, we observed that the disease CGNs for inhibitory and excitatory neurons displayed notably lower network compactness scores compared to their reference counterparts (i.e., loss-of-compactness). This significant decrease in network compactness within the diseased neurons points to a loss of connections among AD-associated genes. Such a loss in the diseased state of inhibitory and excitatory neurons could be a critical factor in the pathogenesis of AD, indicating a disruption in the intricate gene networks that underlie normal neuronal function.

We then focused on prioritizing genes for AD by either differential expression or differential hubness between healthy and diseased states across each cell type (S7 Table). In alignment with the identified cell type specificity for AD, both inhibitory and excitatory neurons demonstrated a more accurate prediction of AD-related genes when analyzed for differential hubness rather than differential expression (Fig 5B, hypergeometric test *P*-value < 0.001). Interestingly, the predictive capacity using differential expression in these neuron types was lower compared to that achieved through differential hubness analysis. Additionally, this capacity was akin to what was observed in other cell types. This finding suggests that a network-based approach is more effective for predicting AD genes than methods solely based on expression, which tend to be less specific to AD-associated cell types. Notably, the overlap between gene predictions made using differential hubness and differential expression was minimal (S5A Fig), indicating that these 2 approaches are complementary to each other in identifying key genes associated with AD.

To delve into the molecular mechanisms implicated in AD pathogenesis within inhibitory and excitatory neurons, we carried out a differential pathway analysis. This analysis was based on CGN signatures of these neurons, focusing on the top 10 genes ranked by hubness (S5B Fig). As anticipated, our analysis revealed that pathways associated with AD and other related neurodegenerative diseases, such as Parkinson's disease and Huntington's disease, were among those most prominently exhibiting a reduced association, or "loss-of-pathway," in inhibitory neurons (Fig 5C). In addition to these, we identified several other pathways that exhibited loss-of-pathway in inhibitory neurons, and these findings were validated through a literature survey. The pathways that were validated to be associated with AD included

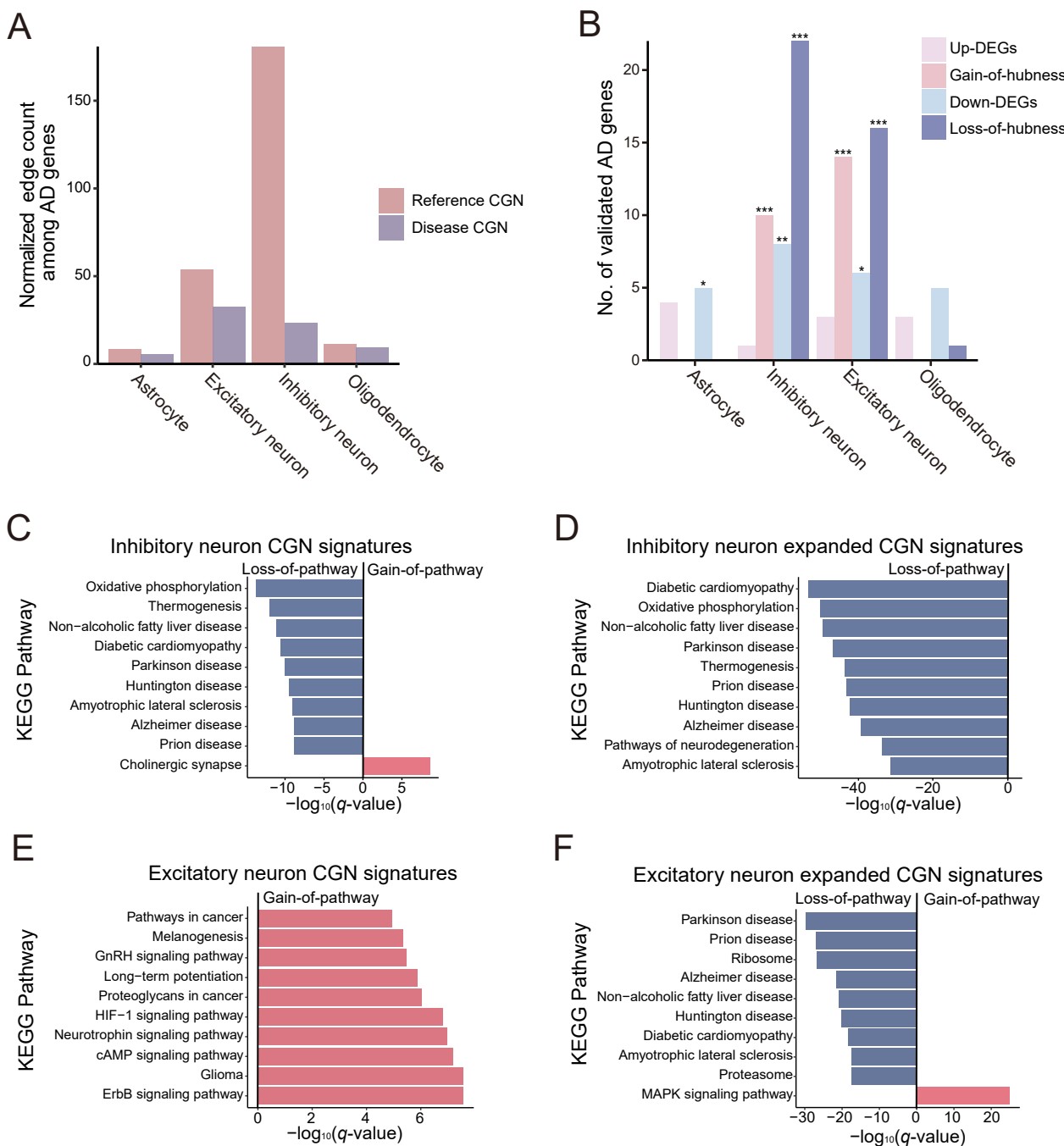

**Fig 5. Cell type-resolved disease genetics for AD.** (**A**) Bar graph depicting the normalized edge count among AD-associated genes in reference and disease CGNs for various neurological cell types. (**B**) Bar graph showing the number of validated AD genes predicted by differential expression or differential hubness across the 4 neurological cell types. Statistical significance of overlap is shown for each gene sets (*$P < 0.05$, ** $P < 0.01$, *** $P < 0.001$ by one-sided hypergeometric test). (**C–F**) Ten most differentially associated KEGG pathways with CGN signature genes: Differentially associated pathways for inhibitory neuron CGN signature (**C**), inhibitory neuron expanded CGN signature (**D**), excitatory neuron CGN signature (**E**), and excitatory neuron expanded CGN signature (**F**). The data underlying this figure can be found in https://doi.org/10.5281/zenodo.14522296. AD, Alzheimer's disease; CGN, cell type-specific gene network.

oxidative phosphorylation [50], thermogenesis [51], non-alcoholic fatty liver disease [52,53], diabetic cardiomyopathy [54,55], amyotrophic lateral sclerosis [56], and prion disease [57]. Significantly, our analysis identified that the pathway related to the cholinergic synapse was the most notably increased in diseased inhibitory neurons. This finding is also relevant given the known involvement of the cholinergic signaling in AD [58]. We also performed our analysis using an expanded CGN signature that includes their network neighbors, which confirmed the initial list of top loss-of-pathways (Fig 5D). This reaffirms the importance of these pathways in the pathogenesis of AD, highlighting their potential roles in the disease's progression and impact within inhibitory neurons.

In our differential pathway analysis using CGN signatures for excitatory neurons, we predominantly observed gain-of-pathways. These are pathways showing increased activity in AD compared to the healthy state, findings which are substantiated by literature evidence (Fig 5E). For instance, the ErbB signaling pathway is known to mediate amyloid-β (Aβ)-induced neurotoxicity [59], and HIF-1 (hypoxia-inducible factor-1) signaling has been found to increase Aβ generation [60]. Additionally, a similar analysis with expanded CGN signatures for excitatory neurons revealed loss-of-pathways akin to those identified in inhibitory neurons (Fig 5F). Among these findings, the MAPK signaling pathway emerged as the most prominent gain-of-pathway. This is in alignment with previous research demonstrating that MKP-1, a crucial negative regulator of MAPKs, can reduce Aβ generation and alleviate cognitive impairments in AD models [61], thereby validating our observation.

## Investigating cell type-resolved lung cancer genetics using HCNetlas

The tumor immune microenvironment has become increasingly recognized as a key hallmark of cancer. Considering this, we hypothesized that HCNetlas CGNs for immune cells could be instrumental in identifying cancer-associated genes that primarily function within the immune microenvironment. Focusing on lung cancer, we constructed disease CGNs for major immune cell types using single-cell transcriptome data derived from tumor tissues of lung cancer patients [28]. Through differential hubness analysis, compared to reference CGNs of corresponding cell types, we pinpointed gain-of-hubness genes predominantly in T cells and myeloid cells, many of which are known to be associated with lung cancer (Fig 6A and S8A Table). Interestingly, only a few gain-of-hubness genes were common across multiple immune cell types, suggesting a specific functional role of cancer-associated genes in T cells and myeloid cells. This analysis also revealed that differential hubness was more effective in identifying lung cancer-associated genes than the traditional differentially expressed genes (DEGs) analysis (Fig 6B and S8B Table). Notably, many up-regulated DEGs shared among all immune cell types included very few validated lung cancer genes. When assessing loss-of-hubness genes, a similar trend was observed: fewer candidates but with more specificity to cell types compared to down-regulated expression in disease (Fig 6C and 6D). T cell-specific loss-of-hubness particularly retrieved a significant number of known lung cancer genes. Additionally, we found supportive literature evidence for the proposed cell type of action for these validated cancer-associated genes identified through differential hubness analysis (S8 Table). This suggests that HCNetlas is effective in predicting genes associated with cancer specifically within immune cell types.

Further evaluation focused on immune checkpoint molecules (ICMs), which are pivotal in antitumor immunity [62,63]. We anticipated an increase in network centrality and expression of ICMs in tumor-derived immune cells. Confirming our hypothesis, genes identified through differential hubness analysis were more effective in detecting ICMs, particularly within T cells and myeloid cells, compared to differential expression analysis (Fig 6E). This finding underscores the advantage of network-based analyses in pinpointing crucial genes in cancer

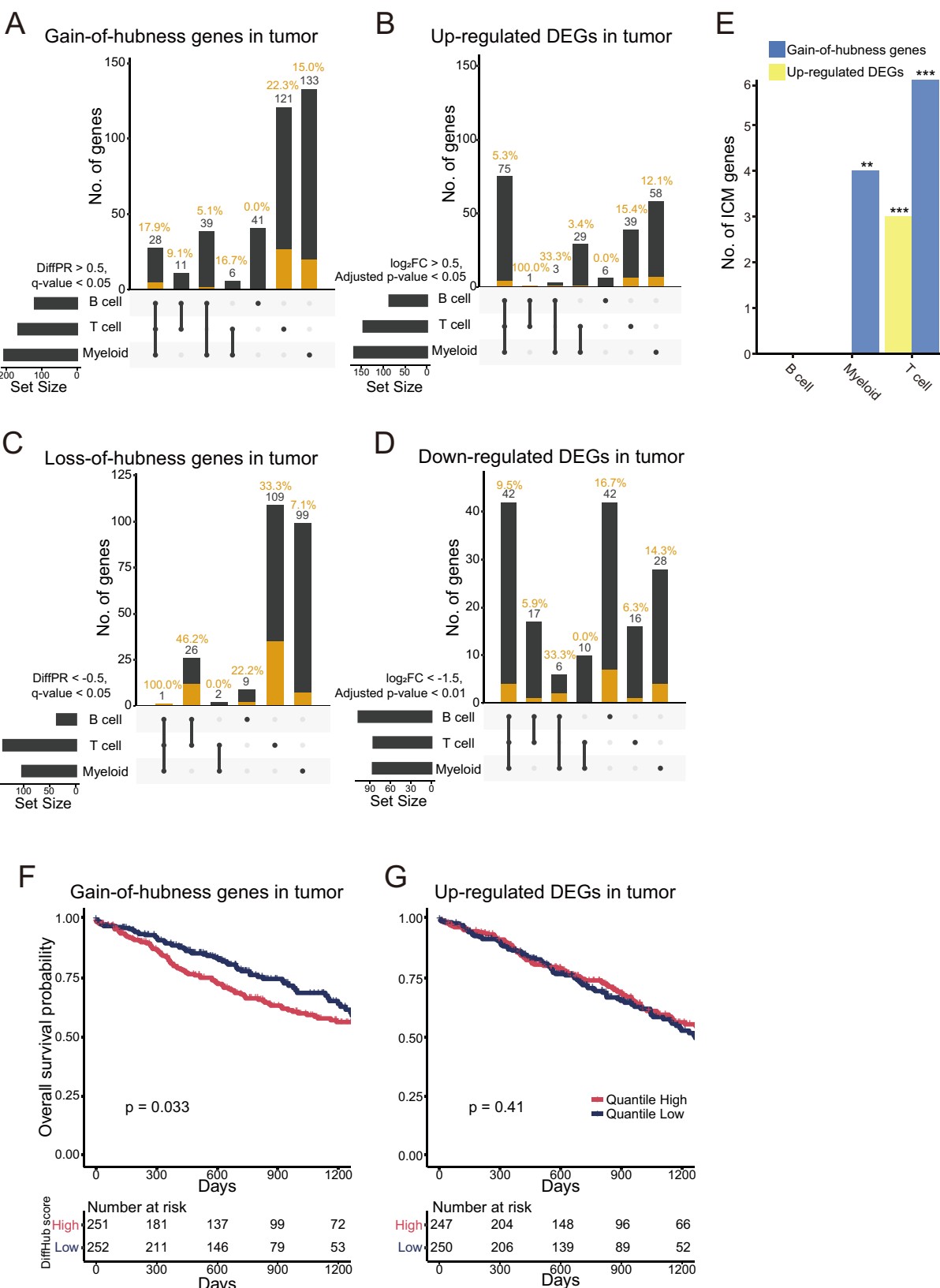

**Fig 6. Identifying genes that contribute to lung cancer through immune cells using HCNetlas.** (**A–D**) UpSet plots for predicted genes for lung cancer by gain-of-hubness (**A**), up-regulated DEGs (**B**), loss-of-hubness (**C**), or down-regulated DEGs (**D**), representing the intersection of predictions across different cell types. Orange bar indicates the number of lung cancer genes validated by various databases such as CancerMine, IntOGen, and cancer gene consensus. (**E**) Bar graph showing the number of ICMs retrieved by both gain-of-hubness genes and up-regulated DEGs across different immune cell types. Statistical significance of overlap is shown for each gene sets (*$P < 0.05$, ** $P < 0.01$, *** $P < 0.001$ by one-sided hypergeometric test). (**F**, **G**) Kaplan–Meier survival curve analysis for cancer patients from TCGA cohort (TCGA-LUSC and TCGA-LUAD), stratified by the enrichment score of gain-of-hubness genes (**F**) or up-regulated DEGs (**G**). The graph shows the overall survival probability over time, with patients categorized into high and low quantiles based on the GSVA score. Significance of survival rate difference between the upper and lower quantile expression groups were evaluated using the log-rank test. The data underlying this figure can be found in https://doi.org/10.5281/zenodo.14522296. DEG, differentially expressed gene; HCNetlas, human cell network atlas; GSVA, gene set variation analysis; ICM, immune checkpoint molecule.

immunology. Additionally, using The Cancer Genome Atlas (TCGA) lung cancer data, we explored the prognostic value of these genes. We found that the gene expression profile association score, calculated using GSVA [19], for the set of gain-of-hubness genes in each tumor sample was predictive of clinical outcomes (Fig 6F), unlike up-regulated DEGs (Fig 6G).

## Discussion

In this study, we have demonstrated the efficacy of a network biology approach for delineating the genetics of disease at the cellular level, making use of HCNetlas—an analytical framework for leveraging compendium of reference CGNs derived from a single-cell expression atlas of healthy individuals for disease research. By comparing these reference CGNs against their diseased counterparts, which are constructed from single-cell transcriptomic data of the same cell types in a disease context, we could measure the alterations in network topology that distinguish healthy from diseased states. We incorporated 3 analytical methods within HCNetlas: differential compactness, differential hubness, and differential pathway analysis. These methods were applied in 3 distinct case studies addressing diseases of the immune system, neurological disorders, and cancer, thereby confirming the extensive applicability of HCNetlas for investigating disease genes in relation to cell type specificity. Our differential compactness analysis pinpointed cell types associated with diseases. We showed that identifying differential hub genes between reference and disease CGNs for a disease-associated cell type is an effective method to predict cell type-specific disease genes. Moreover, by examining differential pathways associated with top hub genes between reference and disease CGNs, we gained insights into the molecular mechanisms potentially driving pathogenesis in the disease-relevant cell types. Consequently, HCNetlas proves to be a robust framework for identifying the specific cell types, genes, and molecular pathways involved in diseases, thus significantly advancing our understanding of how diseases manifest in a cell type-specific manner. Notably, while our work was under review, CellNetdb [64], a database of CGNs across 44 tumor types, was published. Although its CGN inference strategy is similar, CellNetdb is specifically designed to investigate cell type-specific cellular and molecular mechanisms focused on cancer. In contrast, HCNetlas serves as a database of reference CGNs for studying a wide variety of human diseases.

Our study underscores the effectiveness of network-based analysis over conventional expression-based methods in discerning the cell type specificity of disease genes. In our lung cancer case study, for example, the finding that only a few gain-of-hubness genes were shared across multiple immune cell types underscores that it is the alterations in network configuration, rather than just changes in gene expression, that more accurately reflect the cell type-specific functions of genes. Further, our findings reveal that differential hubness offers greater predictive capacity for cancer-associated genes compared to differential expression analysis. A noteworthy observation was that while numerous genes were differentially expressed across

various immune cell types, only a limited subset of these genes were validated to be involved in lung cancer. This highlights that gene properties unique to a cell type, such as differential hubness, can significantly enhance the accuracy of disease gene prediction. Additionally, even though they were not identified through expression-level prioritization, the association of gain-of-hubness genes with expression profiles of tumor samples was found to have prognostic value, unlike the up-regulated DEGs. This implies that the expression levels of genes that influence disease through interactions with other genes in specific cell types are more relevant and indicative of the disease context. Thus, our approach not only identifies disease-relevant genes but also provides insights into the functional significance of these genes within specific cellular environments.

Despite the promising findings, HCNetlas has some limitations. A significant limitation is the current scarcity of "control" single-cell gene expression data for a broad spectrum of cell types and tissues. This lack of data limits the scope and applicability of HCNetlas, as comprehensive mapping of CGNs is contingent on the availability of extensive transcriptomic data. Consequently, our endeavor to create compendium of true reference CGNs was limited by the availability of atlas level resources with varying health conditions (not necessarily diseased). However, this limitation is expected to diminish as the field of single-cell transcriptomics continues to grow. As more data are generated, particularly for healthy tissues, it will become feasible to construct a more comprehensive array of CGNs, covering a wider variety of cell types. Moreover, healthy control samples may vary in properties such as gender, age, body mass index, smoking status, diet, and other lifestyle factors. The effects of these variables will be mitigated by including cells from a large pool of healthy donors, leaving only the core gene network that represent the given cell type within the range of a healthy state. Consequently, the progression of the HCA project is likely to significantly enhance the utility of HCNetlas, extending its applicability to a broader range of diseases and deepening our understanding of cellular behaviors in various pathological states.

Another challenge with HCNetlas stems from the inherent limitations of our network inference methodology, which is dependent on a reference interactome. The reference interactome is mapped predominantly using data from a control state, rather than from a disease state. Consequently, this approach may overlook interactions that are unique to the disease state, as these might be underrepresented or entirely absent in the reference interactome. Such omissions can limit the analytical capacity of HCNetlas, particularly in accurately portraying disease-specific network dynamics. Moreover, low number of cells (below approximately 1,000 cells) often models incomplete network structure, and thus may hinder disease analysis depending on the data input. To avoid this issue, we constructed CGNs only from data sets containing at least 1,000 cells. However, this approach may exclude some important cell types from the database. For example, this has prevented us from observing microglia with our input AD scRNA-seq data, an important cell type known to be associated with the disease. To address this issue, future developments of HCNetlas may need to include the de novo inference of gene networks directly from disease sample data. Integrating these disease-specific networks into HCNetlas would provide a more comprehensive view of the gene interactions occurring in various diseases. This enhancement would not only overcome the current limitations but also enrich the platform capability to provide more nuanced and accurate insights into disease mechanisms at the molecular level.

## Supporting information

**S1 Fig. Overview of cell-type-specific networks (CGNs) of the HCNetlas.** (**A**) Scatter plot depicting the relationship between the number of cells used for network inference (cell count) and network size by node or edge count of CGNs. (**B**) Uniform Manifold Approximation and

Projection (UMAP) visualization of T cell CGNs based on the network node profiles, where each circle's size represents the number of network genes and different colors correspond to various T cell subtypes. (C) UMAP plot displaying T cell CGNs across different organs and tissues. The shapes of the points distinguish between different organs, while the color denotes the specific tissue types. The size of each point corresponds to the size of CGN, as determined by the number of genes. The data underlying this figure can be found in https://doi.org/10.5281/zenodo.14522296.
(PDF)

**S2 Fig. Evaluation of cell type-specific functionality of HCNetlas. (A)** Box plot depicting the interconnectivity distribution of B cell and T cell marker genes within CGNs of the HCNetlas. Marker genes for each cell type were derived from gene sets in the Azimuth and CellMarker databases. (**B**) Heat map displaying the relative percentile ranks of the top 15 hub genes across multiple cell types and tissues, with values scaled per row and color intensity indicating the expression level from low (blue) to high (red). The data underlying this figure can be found in https://doi.org/10.5281/zenodo.14522296.
(PDF)

**S3 Fig. Overlap of cell-type-specific network (CGN) nodes across tissues.** Upset plots illustrate the intersection of network genes across CGNs for various tissues within each cell type. The data underlying this figure can be found in https://doi.org/10.5281/zenodo.14522296.
(PDF)

**S4 Fig. Evaluation of HCNetlas for cell type-resolved disease genetics. (A)** Heatmap displaying the disease profiles of various cell types across different tissues, conducted with gene set variation analysis (GSVA). Each column represents a CGN of HCNetlas, while each row corresponds to a disease gene set sourced from either DisGeNET or GWAS Catalog. Color intensity indicates the degree of association of the CGN signature genes with each disease gene set. (**B**, **C**) Bar graphs showing the within-group connectivity of genes associated with Schizophrenia (B) and Alzheimer's Disease (C) across different cell lineages or tissues. These disease-associated genes were collected from GWAS catalog. The data underlying this figure can be found in https://doi.org/10.5281/zenodo.14522296.
(PDF)

**S5 Fig. Evaluation of HCNetlas for cell-type resolved disease genetics for Alzheimer's disease (AD). (A)** Venn diagram displaying overlap between AD-associated genes predicted by differential hubness and differential expression. (**B**) Networks of signature genes by top 10 hub genes for reference cell type-specific gene network (CGN) from HCNetlas and disease CGNs from disease samples for inhibitory neuron or excitatory neuron. The data underlying this figure can be found in https://doi.org/10.5281/zenodo.14522296.
(PDF)

**S1 Table. HCNetlas cell type, abbreviation, and cell count.**
(XLSX)

**S2 Table. SLE susceptible gene list.** The 184 SLE-associated genes are obtained from KEGG pathway (hsa05322) and KEGG disease (H00080) databases.
(XLSX)

**S3 Table. The list of interferon-stimulated genes.** We compiled interferon-stimulated genes (ISGs) from hallmark gene sets of the molecular signature database (MSigDB) and the Immunological Genome Project (ImmGen) [24], resulting in a total of 423 ISGs.
(XLSX)

**S4 Table. Area under ROC for prediction of interferon-stimulated genes (ISGs) by network centrality.** AUROC were computed with 423 ISGs from S3 Table were applied to genes sorted based on degree centrality.
(XLSX)

**S5 Table. Gain-of-hubness genes and up-regulated genes in SLE myeloid network.** Gain-of-hubness genes were defined by differential percentile rank > 0.5 and $q$-value < 0.05. DEGs were genes with an adjusted $p$-value < 0.05 and a $\log_2$-fold change > 0.5, focusing solely on coding genes.
(XLSX)

**S6 Table. Genes associated with Alzheimer's disease.** AD-associated genes were obtained from the KEGG pathway (M16024), MSigDB (M35868), and Wightman and colleagues.
(XLSX)

**S7 Table. Gain-of-hubness genes, loss-of-hubness genes, up-regulated DEGs, and down-regulated DEGs in AD CGNs.** Gain/loss-of-hubness genes were defined by absolute differential percentile rank > 0.5 and $q$-value < 0.05. DEGs were genes with an adjusted $p$-value < 0.05 and an absolute $\log_2$-fold change > 0.5, focusing solely on coding genes.
(XLSX)

**S8 Table. Gain-of-hubness genes, loss-of-hubness genes, up-regulated DEGs, and down-regulated DEGs in AD CGNs.** Gain/loss-of-hubness genes were defined by absolute differential percentile rank > 0.5 and $q$-value < 0.05. Up-regulated DEGs were genes with an adjusted $p$-value < 0.05 and $\log_2$-fold change > 0.5, and down-regulated DEGs were genes with adjusted $p$-value < 0.01 and $\log_2$-fold change < -1.5.
(XLSX)

## Author Contributions

**Conceptualization:** Jiwon Yu, Junha Cha, Insuk Lee.

**Data curation:** Geon Koh.

**Formal analysis:** Jiwon Yu, Junha Cha.

**Funding acquisition:** Insuk Lee.

**Investigation:** Jiwon Yu, Junha Cha, Insuk Lee.

**Project administration:** Insuk Lee.

**Resources:** Jiwon Yu, Geon Koh.

**Software:** Jiwon Yu.

**Supervision:** Insuk Lee.

**Visualization:** Jiwon Yu.

**Writing – original draft:** Jiwon Yu, Junha Cha.

**Writing – review & editing:** Insuk Lee.

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
