## [Editor Report · Decision Letter 0]

6 Jun 2024

Dear Dr Lee, 

Thank you for submitting your manuscript entitled "HCNetlas: Human cell network atlas enabling cell type-resolved disease genetics" for consideration as a Methods and Resources by PLOS Biology. Please accept my apologies for the unusual delay incurred while we sought external expert advice. 

Your manuscript has now been evaluated by the PLOS Biology editorial staff as well as by an academic editor with relevant expertise and I am writing to let you know that we would like to send your submission out for external peer review.

Once your full submission is complete, your paper will undergo a series of checks in preparation for peer review. After your manuscript has passed the checks it will be sent out for review. To provide the metadata for your submission, please Login to Editorial Manager (https://www.editorialmanager.com/pbiology) within two working days, i.e. by Jun 08 2024 11:59PM.

Kind regards,

Suzanne

Suzanne De Bruijn, PhD, 

Associate Editor

PLOS Biology

sbruijn@plos.org

---

## [Decision Letter · Decision Letter 1]

4 Sep 2024

Dear Dr Lee,

Thank you for your patience while your manuscript "HCNetlas: Human cell network atlas enabling cell type-resolved disease genetics" was peer-reviewed at PLOS Biology as a Methods and Resources Article. Please accept my sincere apologies for the long delays that you have experienced during the peer review process. Your manuscript has now been evaluated by the PLOS Biology editors, an Academic Editor with relevant expertise, and by three independent reviewers. 

In light of the reviews, which you will find at the end of this email, we would like to invite you to revise the work to thoroughly address the reviewers' reports.

As you will see, the reviewers are generally positive about your HCNetlas resource but raise some overlapping concerns with its general utility. Specifically, Reviewer’s #2 and #3 note that the resource has a limited scope since it is based on only two datasets and that this restricts its applicability as a control resource for disease-related gene analysis. They ask that a revised version expands the database to show that it can replace matched control samples as claimed. In addition, Reviewer #1 raises concerns with the number of cells/samples used for the analyses and asks whether multiple samples were used to construct the CGN’s. After discussions with the Academic Editor, we ask that you please comprehensively address the questions that Reviewer #1 raises. 

Given the extent of revision needed, we cannot make a decision about publication until we have seen the revised manuscript and your response to the reviewers' comments. Your revised manuscript is likely to be sent for further evaluation by all or a subset of the reviewers.

**IMPORTANT - SUBMITTING YOUR REVISION**

*Re-submission Checklist*

*Published Peer Review*

*PLOS Data Policy*

*Blot and Gel Data Policy*

Sincerely,

Richard

Richard Hodge, PhD

rhodge@plos.org

REVIEWS:

Reviewer #1: Summary of the paper: 

The authors introduced HCNetlas (human cell network atlas), a collection of reference cell type-specific gene networks (CGNs) from a wide array of healthy tissue cells, and used them as the reference network to compare with disease CGNs. For the comparisons of disease vs healthy CGNs, they developed three network analysis methods, differential compactness, differential hubness, and differential pathways, to investigate cell type-specific functions of disease genes. They applied their methods to systemic lupus erythematosus (SLE), Alzheimer's, and lung cancer disease tissue scRNA-seq samples to identify disease-associated cell types, genes, and pathways. 

The novelty of this paper lies in the development of reference CGNs and their application to disease data to reveal the cell types most affected by diseases. This could facilitate the development of cell-type-resolved diagnostics and therapeutic strategies for complex human diseases. Overall, the paper is original and of good quality, but needs to address the following questions.

Major Comments:

Q1: An individual's disease network consists of both disease-related gene and network alterations and the individual's biological/genetic differences. Using multiple individual samples may not necessarily eliminate all the biological bias. The paper mentioned 1000 single cells as the cutoff for constructing the CGNS. It is not clear whether the paper used multiple samples to construct the CGNs (both healthy and disease). How many samples are needed to produce a robust network? Some discussion on this will be helpful. 

Q2: In evaluating the cell-type-specificity of CGNs (Method), the healthy CGNs are constructed using scHumanNet, a program developed in their previous paper. Please give some basic introduction to the scHumanNet method so readers do not have to go back to read the previous paper.

Q3: (Method) They tested the correctness of CGNs by assuming that genes functionally connected within a CGN reflect the properties of their respective cell type (defined by the GO annotation). They considered the interconnectivity within genes for each cell type as a measure of the cell-type specificity. This is true, but how can this prove the correctness of edges (or each edge) in CGNs? They identified hub genes within each CGN and ranked them using the FindAllHub() function of the scHumanNet. Again, how to prove the correctness of these hub genes?

Q4: (Results) HCNetlas. Figure 1C has many tissue abbreviations. Explanations of these tissues should be displayed somewhere (moved from the supplementary tables to the main text). Why does the CGN network size reach the plateau when the cell count reaches 1000? Why does the network size shrink when the cell count goes over 1000? The network gene expression profiles of the same cell types tend to cluster together in Figure 1B. Aren't the cell types defined by gene expression profiles? Figure 1B can only confirm that the program chose a similar set of genes for the same cell types, but can't prove whether these genes are related to the cell type or not. Can you explain these?

Q5: (Results) HCNetlas as a tool for unraveling cell type specificity of disease genes. They ranked genes in each CNG by network degree centrality and then conducted gene set enrichment analysis using disease-associated genes obtained from DisGeNET and GWAS. For Figure 3A, how are the disease gene sets (rows) ordered? Are they ordered randomly? How are the CGNs (columns) ordered hierarchically? What do the colors/numbers represent? The conclusions drawn from Figure 3A are confusing.

Q6: (Results) What are the definitions of gain and loss of pathways? A pathway consists of multiple genes. How to compute them?

Q7: (Results) How many disease samples were used in the SLE, AD, and Lung Cancer data analysis?

Q8: I found a related paper in the following: Li, Z., Liu, G., Yang, X. et al. An atlas of cell-type-specific interactome networks across 44 human tumor types. Genome Med 16, 30 (2024). https://doi.org/10.1186/s13073-024-01303-w

Reviewer #2: In their manuscript Yu et al describe the setting of a new resource database for the collection of cell type-specific gene networks (CGNs) within specific cell types of various tissues. They want to establish reference CGNs, and then for any given set of disease genes be able to compare to the reference set of CGNs. Thus, the idea behind their project is to make available for researchers a resource of control CGNs that could be used for disease-related gene analysis instead of using matched control samples. Overall, the paper is well-written and include validation with the use of specific disease for assessing differences in CGNs.

Although the idea behind the study is interesting and valuable, I am unsure if at this stage with its application to few datasets, if HCNetlas brings enough for investigators to use it as their controls and what are the plans for extension. Why did the authors limit their CGNs to a few datasets, instead of applying it to all control datasets available in the HCA? It seems that for now HCNetlas is limited to data from 2 published papers, comprising in total 25 organs and the associated 61 cell types identified in these 2 reports. One of these papers is mostly limited to immune cell identification, and thus limiting the whole human being aspect of the analyses and CGNs identified. At this stage, it seems a better fit for any immune-related disorder as most of the cell types included for CGNs are immune cells. 

Unless I am misunderstanding, I am unsure why the authors limited their approach to 2 datasets, albeit 2 very large datasets. If the goal is to eliminate running control samples, this approach would need to be widened and seems limited for now as a general database. It is well-known and described by several sc studies performed on human subjects that the donor-donor variability is huge. So, the limitation here seems to reduce the catalog of 'control samples' to a limited number of donors, 28 which may definitely not be enough to cover all ages for both sexes. Is the goal to extend to all available control datasets deposited in the HCA? This may need to be discussed and added as limitations.

At different parts in the paper, the authors mention using the studies from Dominguez Conde 2022 and the Tabula Sapiens from 2022, as well as the Allen Brain atlas data. However, in the text, they mention using HCA data. Could it be clarified for readers?

A minimum of 1,000 cells for a given cell type for CGN identification may remove from analysis some important cell type that may be related to diseases. This is a major limitation of the study.

On a more philosophical point, there are different factors that could alter gene expression and thus CGNs in human cell types that would not be disease-related but may affect disease-related genes. Or they may be specific conditions that may later CGNs without causing a specific disease. Every human is unique, and may have unique sets of CGNs at distinct point in time, that may vary with aging, although not all of these people will present with a specific disease. How is this taken into account in HCNetlas? What are considered controls from non-controls? Do you plan on applying specific threshold for inclusion in your database, like age, BMI, smoker or not, etc…. Healthy for one cell type may not be healthy for another, and the way control tissues are handled at biobank is limited in terms of metadata. For example, obtaining a healthy tissue, may just mean that this tissue did not present with a tumor, but not that the person in which this tissue was derived was healthy, and thus the CGN may be affected. With such variability how can a reference atlas be handled? 

Reviewer #3: The authors created and tested co-expression based gene-gene networks for human immune and brain cells. They used these networks to identify disruption of specific cell types based on patient samples from Alzheimer's disease and systemic lupus erythematosus. 

Given the word 'atlas' in the title, I assumed this approach would highlight specific cell types body-wide for a given set of disease-associated genes. Instead, this is limited to immune cells from several organs and the brain cell types (which didn't cover the resident immune cells). In this regard, readers might need clarification on how comprehensive this network atlas is. 

GCNs created from the control samples from the SLE and Alzheimer's disease datasets were not compared to results from the HCNetlas GCNs. This needs to be tested since the authors claim their approach can 'circumvent the need for matched control samples.' This may highlight that the AD study took samples from the prefrontal cortex while the HCNetlas GCN is derived from the motor cortex, and was also enriched for neurons. This was done for the lung cancer study, but the conclusion that 'lung cancer genes exert their roles in immune cells' doesn't seem supported, given that the HCNetlas GCN was only from immune cells. 

Figure 2 suggests that the GCNs are partially driven by expression level, given the high connectivity of the B and T cell marker genes. That leaves the question of whether the 'mere changes in expression levels' would highlight the same cell types. 

For Alzheimer's disease, the GWAS-associated genes are expressed highly in microglia cells in the brain, which were not included in HCNetlas. The authors did not find this because they lacked the microglial cells to generate an GCN. In addition, the Alzheimer's GWAS hits are enriched for expression in the spleen and macrophages. This links the peripheral immune system to the disease. These associations were found from body-wide atlases like GTEx, but it doesn't seem HCNetlas would highlight macrophages. So, given its current setup, it would have missed this important characterization of the Alzheimer's disease genes.

Methods 

-were the author-provided cell type annotations used? It seems that they were not and CellTypist was used instead. This should be clearly stated and justified, as the author-provided annotations should be trusted over CellTypist predictions. In addition, the CellTypist parameters should be specified. 

Minor

-for Figure 2C, please mark which genes are within the top15 of the specific four cell types. It seems there is some overlap since only 50 genes are shown. 

-Figure 2 uses GO BP, but KEGG is used later on. GO is more frequently updated and comprehensive and should be used over KEGG. 

-the source of the disease-associated gene sets is highly variable. DisGeNET and KEGG collects genes from many sources with different degrees of evidence. This is clear with the AD genes as 17% are NADH:Ubiquinone Oxidoreductases, which lack genetic evidence and probably end up in the list because KEGG links them together.

---

## [Decision Letter · Decision Letter 2]

11 Dec 2024

Dear Dr Lee,

Thank you for your continued patience while we considered your revised manuscript "HCNetlas: Human cell network atlas enabling cell type-resolved disease genetics" for publication as a Methods and Resources Article at PLOS Biology. Please accept my sincere apologies for the delays that you have experienced during this round of the peer review process. This revised version of your manuscript has been evaluated by the PLOS Biology editors, the Academic Editor and two of the original reviewers.

Based on the reviews, I am pleased to say that we are likely to accept this manuscript for publication, provided you satisfactorily address the following data and other policy-related requests that I have provided below (A-E):

(A) We routinely suggest changes to titles to ensure maximum accessibility for a broad, non-specialist readership. In this case, we would suggest a minor edit to the title, as follows. Please ensure you change both the manuscript file and the online submission system, as they need to match for final acceptance:

“HCNetlas: a reference database of human cell-type-specific gene networks to aid disease genetic analyses”

(B) You may be aware of the PLOS Data Policy, which requires that all data be made available without restriction: http://journals.plos.org/plosbiology/s/data-availability. For more information, please also see this editorial: http://dx.doi.org/10.1371/journal.pbio.1001797

-Supplementary files (e.g., excel). Please ensure that all data files are uploaded as 'Supporting Information' and are invariably referred to (in the manuscript, figure legends, and the Description field when uploading your files) using the following format verbatim: S1 Data, S2 Data, etc. Multiple panels of a single or even several figures can be included as multiple sheets in one excel file that is saved using exactly the following convention: S1_Data.xlsx (using an underscore).

-Deposition in a publicly available repository. Please also provide the accession code or a reviewer link so that we may view your data before publication. 

Figure 1B-C, 2A-E, 3A-B, 4A-F, 5A-F, 6A-G, S1A-C, S2A-B, S3, S4A-C, S5

(C) Please also ensure that each of the relevant figure legends in your manuscript include information on *WHERE THE UNDERLYING DATA CAN BE FOUND*, and ensure your supplemental data file/s has a legend.

(D) Please ensure that your Data Statement in the submission system accurately describes where your data can be found and is in final format, as it will be published as written there. 

(E) Please note that we cannot accept sole deposition of code in GitHub, as this could be changed after publication. However, you can archive this version of your publicly available GitHub code to Zenodo. Once you do this, it will generate a DOI number, which you will need to provide in the Data Accessibility Statement (you are welcome to also provide the GitHub access information). See the process for doing this here: https://docs.github.com/en/repositories/archiving-a-github-repository/referencing-and-citing-content

We expect to receive your revised manuscript within two weeks. 

*Published Peer Review History*

*Press*

Best regards,

Richard

Richard Hodge, PhD

rhodge@plos.org

Reviewer remarks:

Reviewer #2: The authors wrote a very thorough rebuttal and answered all my comments. They modified their text and figures accordingly. I do not have additional comments or questions. 

Reviewer #3: I believe the authors have reasonably addressed the issues I raised.

---

## [Editor Report · Decision Letter 3]

20 Dec 2024

Dear Dr Lee,

On behalf of my colleagues and the Academic Editor, Sui Huang, I am pleased to say that we can accept your manuscript for publication, provided you address any remaining formatting and reporting issues. These will be detailed in an email you should receive within 2-3 business days from our colleagues in the journal operations team; no action is required from you until then. Please note that we will not be able to formally accept your manuscript and schedule it for publication until you have completed any requested changes.

PRESS

Best wishes, 

Richard

Richard Hodge, PhD

rhodge@plos.org

PLOS
